# SchNet: A continuous-filter convolutional neural network for modeling quantum interactions

**K. T. Schütt**[1]*, **P.-J. Kindermans**[1], **H. E. Sauceda**[2], **S. Chmiela**[1]
**A. Tkatchenko**[3], **K.-R. Müller**[1,4,5]†
[1] Machine Learning Group, Technische Universität Berlin, Germany
[2] Theory Department, Fritz-Haber-Institut der Max-Planck-Gesellschaft, Berlin, Germany
[3] Physics and Materials Science Research Unit, University of Luxembourg, Luxembourg
[4] Max-Planck-Institut für Informatik, Saarbrücken, Germany
[5] Dept. of Brain and Cognitive Engineering, Korea University, Seoul, South Korea
* kristof.schuett@tu-berlin.de † klaus-robert.mueller@tu-berlin.de

## Abstract

Deep learning has the potential to revolutionize quantum chemistry as it is ideally suited to learn representations for structured data and speed up the exploration of chemical space. While convolutional neural networks have proven to be the first choice for images, audio and video data, the atoms in molecules are not restricted to a grid. Instead, their precise locations contain essential physical information, that would get lost if discretized. Thus, we propose to use *continuous-filter convolutional layers* to be able to model local correlations without requiring the data to lie on a grid. We apply those layers in SchNet: a novel deep learning architecture modeling quantum interactions in molecules. We obtain a joint model for the total energy and interatomic forces that follows fundamental quantum-chemical principles. Our architecture achieves state-of-the-art performance for benchmarks of equilibrium molecules and molecular dynamics trajectories. Finally, we introduce a more challenging benchmark with chemical and structural variations that suggests the path for further work.

## 1 Introduction

The discovery of novel molecules and materials with desired properties is crucial for applications such as batteries, catalysis and drug design. However, the vastness of chemical compound space and the computational cost of accurate quantum-chemical calculations prevent an exhaustive exploration. In recent years, there have been increased efforts to use machine learning for the accelerated discovery of molecules and materials with desired properties [1–9]. However, these methods are only applied to stable systems in so-called *equilibrium*, i.e., local minima of the potential energy surface $E(\mathbf{r}_1, \ldots, \mathbf{r}_n)$ where $\mathbf{r}_i$ is the position of atom $i$. Data sets such as the established QM9 benchmark [10] contain only equilibrium molecules. Predicting stable atom arrangements is in itself an important challenge in quantum chemistry and material science.

In general, it is *not* clear how to obtain equilibrium conformations without optimizing the atom positions. Therefore, we need to compute both the total energy $E(\mathbf{r}_1, \ldots, \mathbf{r}_n)$ and the forces acting on the atoms

$$\mathbf{F}_i(\mathbf{r}_1, \ldots, \mathbf{r}_n) = -\frac{\partial E}{\partial \mathbf{r}_i}(\mathbf{r}_1, \ldots, \mathbf{r}_n). \tag{1}$$

One possibility is to use a less computationally costly, however, also less accurate quantum-chemical approximation. Instead, we choose to extend the domain of our machine learning model to both compositional (chemical) and configurational (structural) degrees of freedom.

In this work, we aim to learn a representation for molecules using equilibrium and non-equilibrium conformations. Such a general representation for atomistic systems should follow fundamental quantum-mechanical principles. Most importantly, the predicted force field has to be curl-free. Otherwise, it would be possible to follow a circular trajectory of atom positions such that the energy keeps increasing, i.e., breaking the law of energy conservation. Furthermore, the potential energy surface as well as its partial derivatives have to be smooth, e.g., in order to be able to perform geometry optimization. Beyond that, it is beneficial that the model incorporates the invariance of the molecular energy with respect to rotation, translation and atom indexing. Being able to model both chemical and conformational variations constitutes an important step towards ML-driven quantum-chemical exploration.

This work provides the following key contributions:

- We propose *continuous-filter convolutional (cfconv)* layers as a means to move beyond grid-bound data such as images or audio towards modeling objects with arbitrary positions such as astronomical observations or atoms in molecules and materials.

- We propose *SchNet*: a neural network specifically designed to respect essential quantum-chemical constraints. In particular, we use the proposed cfconv layers in $\mathbb{R}^3$ to model interactions of atoms at arbitrary positions in the molecule. SchNet delivers both rotationally invariant energy prediction and rotationally equivariant force predictions. We obtain a smooth potential energy surface and the resulting force-field is guaranteed to be energy-conserving.

- We present a new, challenging benchmark – ISO17 – including both chemical and conformational changes[3]. We show that training with forces improves generalization in this setting as well.

## 2  Related work

Previous work has used neural networks and Gaussian processes applied to hand-crafted features to fit potential energy surfaces [11–16]. Graph convolutional networks for circular fingerprint [17] and molecular graph convolutions [18] learn representations for molecules of arbitrary size. They encode the molecular structure using neighborhood relationships as well as bond features, e.g., one-hot encodings of single, double and triple bonds. In the following, we briefly review the related work that will be used in our empirical evaluation: gradient domain machine learning (GDML), deep tensor neural networks (DTNN) and enn-s2s.

**Gradient-domain machine learning (GDML)**   Chmiela et al. [19] proposed GDML as a method to construct force fields that explicitly obey the law of energy conservation. GDML captures the relationship between energy and interatomic forces (see Eq. 1) by training the gradient of the energy estimator. The functional relationship between atomic coordinates and interatomic forces is thus learned directly and energy predictions are obtained by re-integration. However, GDML does not scale well due to its kernel matrix growing quadratically with the number of atoms as well as the number of examples. Beyond that, it is not designed to represent different compositions of atom types unlike SchNet, DTNN and enn-s2s.

**Deep tensor neural networks (DTNN)**   Schütt et al. [20] proposed the DTNN for molecules that are inspired by the many-body Hamiltonian applied to the interactions of atoms. They have been shown to reach chemical accuracy on a small set of molecular dynamics trajectories as well as QM9. Even though the DTNN shares the invariances with our proposed architecture, its interaction layers lack the continuous-filter convolution interpretation. It falls behind in accuracy compared to SchNet and enn-s2s.

**enn-s2s**   Gilmer et al. [21] proposed the enn-s2s as a variant of message-passing neural networks that uses bond type features in addition to interatomic distances. It achieves state-of-the-art performance on all properties of the QM9 benchmark [21]. Unfortunately, it cannot be used for molecular dynamics predictions (MD-17). This is caused by discontinuities in their potential energy surface due to the

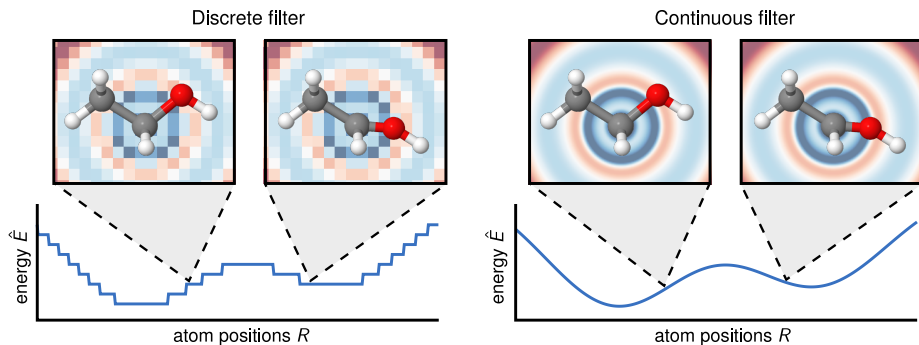

Figure 1: The discrete filter (left) is not able to capture the subtle positional changes of the atoms resulting in discontinuous energy predictions $\hat{E}$ (bottom left). The continuous filter captures these changes and yields smooth energy predictions (bottom right).

discreteness of the one-hot encodings in their input. In contrast, SchNet does not use such features and yields a continuous potential energy surface by using continuous-filter convolutional layers.

## 3 Continuous-filter convolutions

In deep learning, convolutional layers operate on discretized signals such as image pixels [22, 23], video frames [24] or digital audio data [25]. While it is sufficient to define the filter on the same grid in these cases, this is not possible for unevenly spaced inputs such as the atom positions of a molecule (see Fig. 1). Other examples include astronomical observations [26], climate data [27] and the financial market [28]. Commonly, this can be solved by a re-sampling approach defining a representation on a grid [7, 29, 30]. However, choosing an appropriate interpolation scheme is a challenge on its own and, possibly, requires a large number of grid points. Therefore, various extensions of convolutional layers even beyond the Euclidean space exist, e.g., for graphs [31, 32] and 3d shapes[33]. Analogously, we propose to use continuous filters that are able to handle unevenly spaced data, in particular, atoms at arbitrary positions.

Given the feature representations of $n$ objects $X^l = (\mathbf{x}_1^l, \ldots, \mathbf{x}_n^l)$ with $\mathbf{x}_i^l \in \mathbb{R}^F$ at locations $R = (\mathbf{r}_1, \ldots, \mathbf{r}_n)$ with $\mathbf{r}_i \in \mathbb{R}^D$, the continuous-filter convolutional layer $l$ requires a filter-generating function

$$W^l : \mathbb{R}^D \rightarrow \mathbb{R}^F,$$

that maps from a position to the corresponding filter values. This constitutes a generalization of a filter tensor in discrete convolutional layers. As in dynamic filter networks [34], this filter-generating function is modeled with a neural network. While dynamic filter networks generate weights restricted to a grid structure, our approach generalizes this to arbitrary position and number of objects. The output $\mathbf{x}_i^{l+1}$ for the convolutional layer at position $\mathbf{r}_i$ is then given by

$$\mathbf{x}_i^{l+1} = (X^l * W^l)_i = \sum_j \mathbf{x}_j^l \circ W^l(\mathbf{r}_i - \mathbf{r}_j), \tag{2}$$

where "$\circ$" represents the element-wise multiplication. We apply these convolutions feature-wise for computational efficiency [35]. The interactions between feature maps are handled by separate object-wise or, specifically, atom-wise layers in SchNet.

## 4 SchNet

SchNet is designed to learn a representation for the prediction of molecular energies and atomic forces. It reflects fundamental physical laws including invariance to atom indexing and translation, a smooth energy prediction w.r.t. atom positions as well as energy-conservation of the predicted force fields. The energy and force predictions are rotationally invariant and equivariant, respectively.

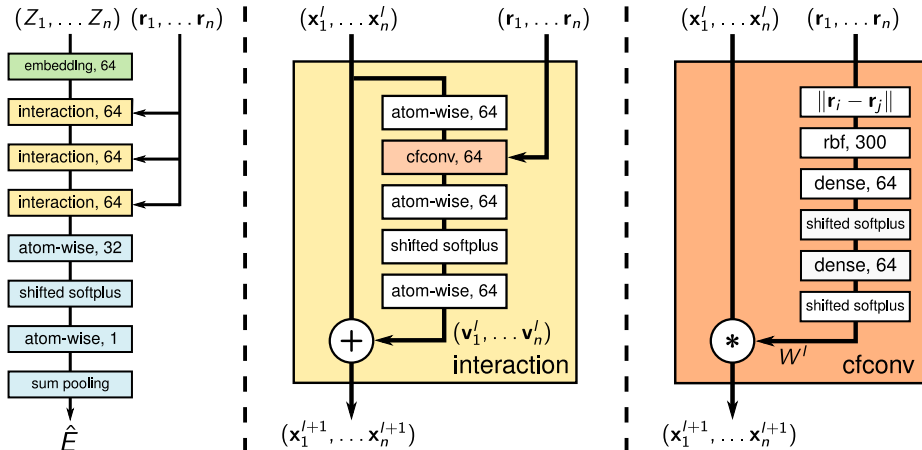

Figure 2: Illustration of SchNet with an architectural overview (left), the interaction block (middle) and the continuous-filter convolution with filter-generating network (right). The shifted softplus is defined as $\mathrm{ssp}(x) = \ln(0.5e^x + 0.5)$.

## 4.1 Architecture

Fig. 2 shows an overview of the SchNet architecture. At each layer, the molecule is represented atom-wise analogous to pixels in an image. Interactions between atoms are modeled by the three interaction blocks. The final prediction is obtained after atom-wise updates of the feature representation and pooling of the resulting atom-wise energy. In the following, we discuss the different components of the network.

**Molecular representation** A molecule in a certain conformation can be described uniquely by a set of $n$ atoms with nuclear charges $Z = (Z_1, \ldots, Z_n)$ and atomic positions $R = (\mathbf{r}_1, \ldots \mathbf{r}_n)$. Through the layers of the neural network, we represent the atoms using a tuple of features $X^l = (\mathbf{x}_1^l, \ldots \mathbf{x}_n^l)$, with $\mathbf{x}_i^l \in \mathbb{R}^F$ with the number of feature maps $F$, the number of atoms $n$ and the current layer $l$. The representation of atom $i$ is initialized using an embedding dependent on the atom type $Z_i$:

$$\mathbf{x}_i^0 = \mathbf{a}_{Z_i}. \tag{3}$$

The atom type embeddings $\mathbf{a}_Z$ are initialized randomly and optimized during training.

**Atom-wise layers** A recurring building block in our architecture are atom-wise layers. These are dense layers that are applied separately to the representation $\mathbf{x}_i^l$ of atom $i$:

$$\mathbf{x}_i^{l+1} = W^l \mathbf{x}_i^l + \mathbf{b}^l$$

These layers is responsible for the recombination of feature maps. Since weights are shared across atoms, our architecture remains scalable with respect to the size of the molecule.

**Interaction** The interaction blocks, as shown in Fig. 2 (middle), are responsible for updating the atomic representations based on the molecular geometry $R = (\mathbf{r}_1, \ldots \mathbf{r}_n)$. We keep the number of feature maps constant at $F = 64$ throughout the interaction part of the network. In contrast to MPNN and DTNN, we do not use weight sharing across multiple interaction blocks.

The blocks use a residual connection inspired by ResNet [36]:

$$\mathbf{x}_i^{l+1} = \mathbf{x}_i^l + \mathbf{v}_i^l.$$

As shown in the interaction block in Fig. 2, the residual $\mathbf{v}_i^l$ is computed through an atom-wise layer, an interatomic continuous-filter convolution (cfconv) followed by two more atom-wise layers with a softplus non-linearity in between. This allows for a flexible residual that incorporates interactions between atoms and feature maps.

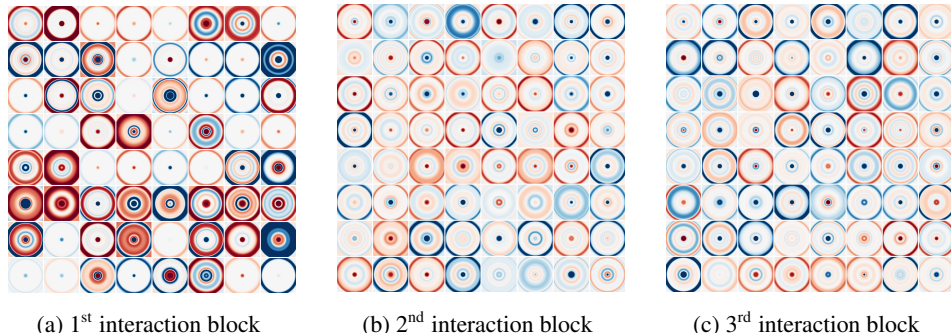

| (a) 1$^{\text{st}}$ interaction block | (b) 2$^{\text{nd}}$ interaction block | (c) 3$^{\text{rd}}$ interaction block |

Figure 3: 10x10 Å cuts through all 64 radial, three-dimensional filters in each interaction block of SchNet trained on molecular dynamics of ethanol. Negative values are blue, positive values are red.

**Filter-generating networks** The cfconv layer including its filter-generating network are depicted at the right panel of Fig. 2. In order to satisfy the requirements for modeling molecular energies, we restrict our filters for the cfconv layers to be rotationally invariant. The rotational invariance is obtained by using interatomic distances

$$d_{ij} = \|\mathbf{r}_i - \mathbf{r}_j\|$$

as input for the filter network. Without further processing, the filters would be highly correlated since a neural network after initialization is close to linear. This leads to a plateau at the beginning of training that is hard to overcome. We avoid this by expanding the distance with radial basis functions

$$e_k(\mathbf{r}_i - \mathbf{r}_j) = \exp(-\gamma \|d_{ij} - \mu_k\|^2)$$

located at centers $0\text{Å} \leq \mu_k \leq 30\text{Å}$ every $0.1\text{Å}$ with $\gamma = 10\text{Å}$. This is chosen such that all distances occurring in the data sets are covered by the filters. Due to this additional non-linearity, the initial filters are less correlated leading to a faster training procedure. Choosing fewer centers corresponds to reducing the resolution of the filter, while restricting the range of the centers corresponds to the filter size in a usual convolutional layer. An extensive evaluation of the impact of these variables is left for future work. We feed the expanded distances into two dense layers with softplus activations to compute the filter weight $W(\mathbf{r}_i - \mathbf{r}_j)$ as shown in Fig. 2 (right).

Fig 3 shows 2d-cuts through generated filters for all three interaction blocks of SchNet trained on an ethanol molecular dynamics trajectory. We observe how each filter emphasizes certain ranges of interatomic distances. This enables its interaction block to update the representations according to the radial environment of each atom. The sequential updates from three interaction blocks allow SchNet to construct highly complex many-body representations in the spirit of DTNNs [20] while keeping rotational invariance due to the radial filters.

## 4.2 Training with energies and forces

As described above, the interatomic forces are related to the molecular energy, so that we can obtain an energy-conserving force model by differentiating the energy model w.r.t. the atom positions

$$\hat{\mathbf{F}}_i(Z_1, \ldots, Z_n, \mathbf{r}_1, \ldots, \mathbf{r}_n) = -\frac{\partial \hat{E}}{\partial \mathbf{r}_i}(Z_1, \ldots, Z_n, \mathbf{r}_1, \ldots, \mathbf{r}_n). \tag{4}$$

Chmiela et al. [19] pointed out that this leads to an energy-conserving force-field by construction. As SchNet yields rotationally invariant energy predictions, the force predictions are rotationally equivariant by construction. The model has to be at least twice differentiable to allow for gradient descent of the force loss. We chose a shifted softplus $\text{ssp}(x) = \ln(0.5e^x + 0.5)$ as non-linearity throughout the network in order to obtain a smooth potential energy surface. The shifting ensures that $\text{ssp}(0) = 0$ and improves the convergence of the network. This activation function shows similarity to ELUs [37], while having infinite order of continuity.

Table 1: Mean absolute errors for energy predictions in kcal/mol on the QM9 data set with given training set size $N$. Best model in bold.

| $N$ | SchNet | DTNN [20] | enn-s2s [21] | enn-s2s-ens5 [21] |
|---|---|---|---|---|
| 50,000 | **0.59** | 0.94 | – | – |
| 100,000 | **0.34** | 0.84 | – | – |
| 110,462 | **0.31** | – | 0.45 | 0.33 |

We include the total energy $E$ as well as forces $\mathbf{F}_i$ in the training loss to train a neural network that performs well on both properties:

$$\ell(\hat{E}, (E, \mathbf{F}_1, \ldots, \mathbf{F}_n)) = \rho\|E - \hat{E}\|^2 + \frac{1}{n}\sum_{i=0}^{n}\left\|\mathbf{F}_i - \left(-\frac{\partial\hat{E}}{\partial\mathbf{R}_i}\right)\right\|^2. \tag{5}$$

This kind of loss has been used before for fitting a restricted potential energy surfaces with MLPs [38]. In our experiments, we use $\rho = 0.01$ for combined energy and force training. The value of $\rho$ was optimized empirically to account for different scales of energy and forces.

Due to the relation of energies and forces reflected in the model, we expect to see improved generalization, however, at a computational cost. As we need to perform a full forward and backward pass on the energy model to obtain the forces, the resulting force model is twice as deep and, hence, requires about twice the amount of computation time.

Even though the GDML model captures this relationship between energies and forces, it is explicitly optimized to predict the force field while the energy prediction is a by-product. Models such as circular fingerprints [17], molecular graph convolutions or message-passing neural networks[21] for property prediction across chemical compound space are only concerned with equilibrium molecules, i.e., the special case where the forces are vanishing. They can not be trained with forces in a similar manner, as they include discontinuities in their predicted potential energy surface caused by discrete binning or the use of one-hot encoded bond type information.

## 5 Experiments and results

In this section, we apply the SchNet to three different quantum chemistry datasets: QM9, MD17 and ISO17. We designed the experiments such that each adds another aspect towards modeling chemical space. While QM9 only contains equilibrium molecules, for MD17 we predict conformational changes of molecular dynamics of single molecules. Finally, we present ISO17 combining both chemical and structural changes.

For all datasets, we report mean absolute errors in kcal/mol for the energies and in kcal/mol/Å for the forces. The architecture of the network was fixed after an evaluation on the MD17 data sets for benzene and ethanol (see supplement). In each experiment, we split the data into a training set of given size $N$ and use a validation set of 1,000 examples for early stopping. The remaining data is used as test set. All models are trained with SGD using the ADAM optimizer [39] with 32 molecules per mini-batch. We use an initial learning rate of $10^{-3}$ and an exponential learning rate decay with ratio 0.96 every 100,000 steps. The model used for testing is obtained using an exponential moving average over weights with decay rate 0.99.

### 5.1 QM9 – chemical degrees of freedom

QM9 is a widely used benchmark for the prediction of various molecular properties in equilibrium [10, 40, 41]. Therefore, the forces are zero by definition and do not need to be predicted. In this setting, we train a single model that generalizes across different compositions and sizes.

QM9 consists of $\approx$130k organic molecules with up to 9 heavy atoms of the types {C, O, N, F}. As the size of the training set varies across previous work, we trained our models each of these experimental settings. Table 1 shows the performance of various competing methods for predicting the total energy (property $U_0$ in QM9). We provide comparisons to the DTNN [20] and the best performing MPNN configuration denoted *enn-s2s* and an ensemble of MPNNs (enn-s2s-ens5) [21]. SchNet consistently obtains state-of-the-art performance with an MAE of 0.31 kcal/mol at 110k training examples.

Table 2: Mean absolute errors for energy and force predictions in kcal/mol and kcal/mol/Å, respectively. GDML and SchNet test errors for training with 1,000 and 50,000 examples of molecular dynamics simulations of small, organic molecules are shown. SchNets were trained only on energies as well as energies and forces combined. Best results in bold.

| | | N = 1,000 | | | N = 50,000 | | |
| | | GDML [19] | SchNet | | DTNN [20] | SchNet | |
| | | forces | energy | both | energy | energy | both |
| **Benzene** | energy | **0.07** | 1.19 | 0.08 | **0.04** | 0.08 | 0.07 |
| | forces | **0.23** | 14.12 | 0.31 | – | 1.23 | **0.17** |
| **Toluene** | energy | **0.12** | 2.95 | **0.12** | 0.18 | 0.16 | **0.09** |
| | forces | **0.24** | 22.31 | 0.57 | – | 1.79 | **0.09** |
| **Malonaldehyde** | energy | 0.16 | 2.03 | **0.13** | 0.19 | 0.13 | **0.08** |
| | forces | 0.80 | 20.41 | **0.66** | – | 1.51 | **0.08** |
| **Salicylic acid** | energy | **0.12** | 3.27 | 0.20 | 0.41 | 0.25 | **0.10** |
| | forces | **0.28** | 23.21 | 0.85 | – | 3.72 | **0.19** |
| **Aspirin** | energy | **0.27** | 4.20 | 0.37 | – | 0.25 | **0.12** |
| | forces | **0.99** | 23.54 | 1.35 | – | 7.36 | **0.33** |
| **Ethanol** | energy | 0.15 | 0.93 | **0.08** | – | 0.07 | **0.05** |
| | forces | 0.79 | 6.56 | **0.39** | – | 0.76 | **0.05** |
| **Uracil** | energy | **0.11** | 2.26 | 0.14 | – | 0.13 | **0.10** |
| | forces | **0.24** | 20.08 | 0.56 | – | 3.28 | **0.11** |
| **Naphtalene** | energy | **0.12** | 3.58 | 0.16 | – | 0.20 | **0.11** |
| | forces | **0.23** | 25.36 | 0.58 | – | 2.58 | **0.11** |

## 5.2 MD17 – conformational degrees of freedom

MD17 is a collection of eight molecular dynamics simulations for small organic molecules. These data sets were introduced by Chmiela et al. [19] for prediction of energy-conserving force fields using GDML. Each of these consists of a trajectory of a single molecule covering a large variety of conformations. Here, the task is to predict energies and forces using a separate model for each trajectory. This molecule-wise training is motivated by the need for highly-accurate force predictions when doing molecular dynamics.

Table 2 shows the performance of SchNet using 1,000 and 50,000 training examples in comparison with GDML and DTNN. Using the smaller data set, GDML achieves remarkably accurate energy and force predictions despite being only trained on forces. The energies are only used to fit the integration constant. As mentioned before, GDML does not scale well with the number of atoms and training examples. Therefore, it cannot be trained on 50,000 examples. The DTNN was evaluated only on four of these MD trajectories using the larger training set [20]. Note that the *enn-s2s* cannot be used on this dataset due to discontinuities in its inferred potential energy surface.

We trained SchNet using just energies and using both energies and forces. While the energy-only model shows high errors for the small training set, the model including forces achieves energy predictions comparable to GDML. In particular, we observe that SchNet outperforms GDML on the more flexible molecules malonaldehyde and ethanol, while GDML reaches much lower force errors on the remaining MD trajectories that all include aromatic rings.

The real strength of SchNet is its scalability, as it outperforms the DTNN in three of four data sets using 50,000 training examples using only energies in training. Including force information, SchNet consistently obtains accurate energies and forces with errors below 0.12 kcal/mol and 0.33 kcal/mol/Å, respectively. Remarkably, when training on energies and forces using 1,000 training examples, SchNet performs better than training the same model on energies alone for 50,000 examples.

Table 3: Mean absolute errors on $C_7O_2H_{10}$ isomers in kcal/mol.

| | | mean predictor | SchNet | |
| --- | --- | --- | --- | --- |
| | | | *energy* | *energy+forces* |
| **known molecules /** | *energy* | 14.89 | 0.52 | **0.36** |
| **unknown conformation** | *forces* | 19.56 | 4.13 | **1.00** |
| **unknown molecules /** | *energy* | 15.54 | 3.11 | **2.40** |
| **unknown conformation** | *forces* | 19.15 | 5.71 | **2.18** |

### 5.3  ISO17 – chemical and conformational degrees of freedom

As the next step towards quantum-chemical exploration, we demonstrate the capability of SchNet to represent a complex potential energy surface including conformational and chemical changes. We present a new dataset – ISO17 – where we consider short MD trajectories of 129 isomers, i.e., chemically different molecules with the same number and types of atoms. In contrast to MD17, we train a joint model across different molecules. We calculate energies and interatomic forces from short MD trajectories of 129 molecules drawn randomly from the largest set of isomers in QM9. While the composition of all included molecules is $C_7O_2H_{10}$, the chemical structures are fundamentally different. With each trajectory consisting of 5,000 conformations, the data set consists of 645,000 labeled examples.

We consider two scenarios with this dataset: In the first variant, the molecular graph structures present in training are also present in the test data. This demonstrates how well our model is able to represent a complex potential energy surface with chemical and conformational changes. In the more challenging scenario, the test data contains a different subset of molecules. Here we evaluate the generalization of our model to previously unseen chemical structures. We predict forces and energies in both cases and compare to the mean predictor as a baseline. We draw a subset of 4,000 steps from 80% of the MD trajectories for training and validation. This leaves us with a separate test set for each scenario: (1) the unseen 1,000 conformations of molecule trajectories included in the training set and (2) all 5,000 conformations of the remaining 20% of molecules not included in training.

Table 3 shows the performance of the SchNet on both test sets. Our proposed model reaches chemical accuracy for the prediction of energies and forces for the test set of known molecules. Including forces in the training improves the performance here as well as on the set of unseen molecules. This shows that using force information does not only help to accurately predict nearby conformations of a single molecule, but indeed helps to generalize across chemical compound space.

## 6  Conclusions

We have proposed continuous-filter convolutional layers as a novel building block for deep neural networks. In contrast to the usual convolutional layers, these can model unevenly spaced data as occurring in astronomy, climate reasearch and, in particular, quantum chemistry. We have developed SchNet to demonstrate the capabilities of continuous-filter convolutional layers in the context of modeling quantum interactions in molecules. Our architecture respects quantum-chemical constraints such as rotationally invariant energy predictions as well as rotationally equivariant, energy-conserving force predictions.

We have evaluated our model in three increasingly challenging experimental settings. Each brings us one step closer to practical chemical exploration driven by machine learning. SchNet improves the state-of-the-art in predicting energies for molecules in equilibrium of the QM9 benchmark. Beyond that, it achieves accurate predictions for energies and forces for all molecular dynamics trajectories in MD17. Finally, we have introduced ISO17 consisting of 645,000 conformations of various $C_7O_2H_{10}$ isomers. While we achieve promising results on this new benchmark, modeling chemical and conformational variations remains difficult and needs further improvement. For this reason, we expect that ISO17 will become a new standard benchmark for modeling quantum interactions with machine learning.

**Acknowledgments**

This work was supported by the Federal Ministry of Education and Research (BMBF) for the Berlin Big Data Center BBDC (01IS14013A). Additional support was provided by the DFG (MU 987/20-1) and from the European Union's Horizon 2020 research and innovation program under the Marie Sklodowska-Curie grant agreement NO 657679. K.R.M. gratefully acknowledges the BK21 program funded by Korean National Research Foundation grant (No. 2012-005741) and the Institute for Information & Communications Technology Promotion (IITP) grant funded by the Korea government (no. 2017-0-00451).

## Footnotes

[3]ISO17 is publicly available at `www.quantum-machine.org`.

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
