[Supplementary Material · supplement.pdf]

# Supplement

## Model selection

We evaluate the performance of SchNet using $T \in \{1, 2, 3\}$ interaction blocks. We also compare to using shared weights across interaction blocks in spirit of transition functions in graph neural networks. Here, we additionally list root mean squared errors (RMSE). For all models with $T \geq 2$, we observe only minor difference in performance. Therefore, we choose SchNet without shared weights across interaction blocks and $T = 3$ throughout this work.

Table 1: Test errors of networks with $T \in \{1, 2, 3\}$ shared and unshared interaction blocks on ethanol and benzene data sets trained on energies and forces using 1k examples.

| | | Ethanol | | | | Benzene | | | |
|---|---|---|---|---|---|---|---|---|---|
| | | *Energy [kcal/mol]* | | *Force [kcal/mol/Å]* | | *Energy [kcal/mol]* | | *Force [kcal/mol/Å]* | |
| *i* | *T* | *MAE* | *RMSE* | *MAE* | *RMSE* | *MAE* | *RMSE* | *MAE* | *RMSE* |
| | 1 | 0.43 | 0.57 | 1.72 | 2.52 | **0.08** | 0.11 | 0.35 | 0.66 |
| | 2 | **0.08** | 0.13 | 0.40 | 0.70 | **0.08** | **0.10** | **0.30** | **0.46** |
| | 3 | **0.08** | 0.14 | **0.39** | 0.72 | **0.08** | **0.10** | 0.31 | 0.47 |
| shared | 2 | **0.08** | 0.16 | 0.43 | 0.82 | **0.08** | **0.10** | 0.31 | 0.52 |
| | 3 | **0.08** | **0.12** | **0.39** | **0.68** | **0.08** | **0.10** | 0.31 | 0.47 |

Table 2: Test errors of networks with $T \in \{1, 2, 3\}$ interaction blocks with shared and unshared filter-generating networks on ethanol and benzene data sets trained on energies and forces using 50k examples.

| | | Ethanol | | | | Benzene | | | |
|---|---|---|---|---|---|---|---|---|---|
| | | *Energy [kcal/mol]* | | *Force [kcal/mol/Å]* | | *Energy [kcal/mol]* | | *Force [kcal/mol/Å]* | |
| *i* | *T* | *MAE* | *RMSE* | *MAE* | *RMSE* | *MAE* | *RMSE* | *MAE* | *RMSE* |
| | 1 | 0.34 | 0.45 | 1.34 | 1.94 | 0.08 | 0.10 | 0.31 | 0.48 |
| | 2 | 0.05 | 0.06 | 0.07 | 0.11 | **0.07** | **0.09** | 0.20 | 0.30 |
| | 3 | **0.05** | **0.06** | **0.05** | **0.08** | **0.07** | **0.09** | **0.17** | **0.27** |
| shared | 2 | **0.05** | **0.06** | 0.09 | 0.14 | 0.08 | 0.10 | 0.23 | 0.35 |
| | 3 | **0.05** | **0.06** | 0.08 | 0.13 | **0.07** | **0.09** | 0.18 | **0.27** |

**ISO-17: $C_7O_2H_{10}$ isomer data set**

Figure 1: Illustration of experimental settings as they appear in the data sets used for evaluation: molecules in equilibrium (QM9), molecular dynamics trajectories of single molecules (MD17) and combination of chemical and structural degrees of freedom (ISO17).

The database was generated from molecular dynamics simulations using the Fritz-Haber Institute ab initio simulation package (FHI-aims)[1]. The simulations were carried out using the standard quantum chemistry computational method density functional theory (DFT) in the generalized gradient approximation (GGA) with the Perdew-Burke-Ernzerhof (PBE) functional[2] and the Tkatchenko-Scheffler (TS) van der Waals correction method[3]. The database consist of 129 molecules each containing 5,000 conformational geometries, energies and forces with a resolution of 1 femtosecond in the molecular dynamics trajectories. The molecules were randomly drawn from the largest set of isomers in the QM9 dataset which consists of molecules with a fixed composition of atoms ($C_7O_2H_{10}$) arranged in different chemically valid structures. The data is publicly available at http://www.quantum-machine.org/datasets/

[1] Blum, V.; Gehrke, R.; Hanke, F.; Havu, P.; Havu, V.; Ren, X.; Reuter, K.; Scheffler, M. Ab Initio Molecular Simulations with Numeric Atom-Centered Orbitals. Comput. Phys. Commun. 2009, 180 (11), 2175–2196.

[2] Perdew, J. P.; Burke, K.; Ernzerhof, M. Generalized Gradient Approximation Made Simple. Phys. Rev. Lett. 1996, 77 (18), 3865–3868.

[3] Tkatchenko, A.; Scheffler, M. Accurate Molecular Van Der Waals Interactions from Ground-State Electron Density and Free-Atom Reference Data. Phys. Rev. Lett. 2009, 102 (7), 73005.