[Reviews · NeurIPS 2017]

Reviewer 1



Summary: In order to design new molecules, one needs to predict if the newly designed molecules could reach an equilibrium state. All possible configurations of atoms do not lead to such state. To evaluate that, the authors propose to learn predictors of such equilibrium from annotated datasets, using convolutional networks. Different to previous works, that might lead to inaccurate predictions from convolutions computed in a discretized space, the authors propose continuous filter convolution layers. They compare their approach with two previous ones on 2 datasets and also introduce a third dataset. Their model works better than the others especially when using more training data. I don't have a very strong background in physics so I'm not the best person to assess the quality of this work, but I found that the proposed solution made sense, the paper is clearly written and seems technically correct. Minor: don't forget to replace the XXX Points 3,4 should not be listed as contributions in my opinion, they are results l. 26 : the both the -> both the Ref 14: remove first names

Reviewer 2



I have no expertise in DL applied to chemistry. I have seen other papers on that topic, but I cannot evaluate what is novel, what is trivial, and what is based on sound principles. I get the impression from the paper that the authors have a good understanding of topic. I would like to ask the authors for a clarification about what they do with their "continuous convolution". It seems to me that they are simply using a kind of radial basis function, with a finite number of locations. Then they use a matrix multiplication followed by element-wise multiplication. It feels more like good old interpolation in the space of locations with a finite number of basis elements. Seems like a nice application, but I don't know much about the chemistry part. Also, I think the authors should mention Residual Networks at some place because their design from Figure 2 really looks like it features Resnet blocks.

Reviewer 3



This paper proposes a new deep learning architecture for modeling the quantum structure of small molecules, including a particular type of layer with a continuous response in the location of the atom nuclei. The authors also combine a number of different ideas to address the particular challenges of this application: “dynamic filters,” graph-convolutions, and supervised targets for the gradient at the input layer. I think this is an interesting paper, both for its significance to chemical physics and its significance to the deep learning community. It combines a number of recently-proposed ideas into a single deep learning architecture. The first two sections could be better organized. For a machine learning audience, it might be useful to clearly list the idiosyncrasies of this application and how they are being modeled in the proposed approach. In particular, the first two paragraphs of Section 2 reference other work without clearly explaining the relationship.